# A Comprehensive Performance Evaluation of Chinese Energy Supply Chain under "Double-Carbon" Goals Based on AHP and Three-Stage DEA

**Xiaoqing Huang [1,2,*], Xiaoyong Lu [3], Yuqi Sun [2,*], Jingui Yao [4] and Wenxing Zhu [2]**

[1]  School of Public Policy and Administration, Nanchang University, Nanchang 330031, China
[2]  School of Business, Jiangxi University of Science and Technology, Nanchang 330013, China
[3]  Gongqing Institute of Science and Technology, Jiujiang 332020, China
[4]  Nanchang JiaoTong Institute, Nanchang 330100, China
[*]  Correspondence: huangxiaoqing2111@163.com (X.H.); sunyuqi1027@163.com (Y.S.)

**Abstract:** In 2020, China put forward the goals of "peak carbon dioxide emissions" and "carbon neutrality" ("double-carbon") and it is urgent for the energy industry to achieve green transformation. Aiming at the rigid requirements of the carbon-peaking and carbon-neutrality goals ("double-carbon"), this study established a performance evaluation index system for an energy supply chain of a four-tier structure based on the "double-carbon" goals, calculating its weight by the analytic hierarchy process (AHP). On this basis, a three-stage data envelopment analysis (DEA) evaluation model was established to evaluate the performance of the energy supply chain in 2010–2019. According to the three-stage DEA evaluation mode, the initial input–output efficiency value of the energy supply chain was calculated by the DEA-BCC (extended by Banker, Charnes and Cooper) model and DEA-CCR (proposed by Charnes, Cooper and Rhodes) model and the influence of environmental noise was eliminated by stochastic frontier analysis (SFA) regression; we then obtained the adjusted efficiency value for the energy supply chain. At the same time, taking 2015 as the dividing point, the advantages and disadvantages between the traditional energy supply chain and new energy supply chain were analyzed and summarized. Further analysis and suggestions are provided to consumers, enterprises and countries from four aspects: energy supply, energy production and processing, energy transmission and distribution and energy consumption.

**Keywords:** "double-carbon"; Chinese energy supply chain; performance evaluation; AHP; three-stage DEA



## 1. Introduction

Because of the major threat to human society posed by global climate change, more and more countries have upgraded "carbon neutrality" to a national strategy and have put forward the vision of a carbon-free future [1]. In 2020, based on the inherent requirements of promoting sustainable development and the responsibility of building a Community of Shared Future for Mankind, China announced the target vision of "peak carbon dioxide emissions" and "carbon neutrality" ("double-carbon") [2]. Under the "double-carbon" goals, as one of the industries with high energy consumption, the traditional energy enterprises that have mainly relied on coal are faced with a severe impact from the substitution of dominant industries and so it is urgent to achieve green transformation. At the same time, this plays a vital role in helping to achieve the "double-carbon" goals in the whole energy supply chain [3–5]. In order to carry out the green and "double-carbon" transformations in the energy industries, in addition to national policy supports and sufficient funds, it is necessary to accurately assess the "double-carbon" and environmental-protection situations of each node in the energy supply chain so as to put forward the transformation strategy [6,7].

On the basis of the traditional supply chain performance evaluation system, this study considered the "double-carbon" goal, increased some relevant indicators, such as carbon emissions, carbon recovery and so on, and improved the integrity and practicability of the supply chain performance evaluation system [8]. The construction of the energy supply chain performance evaluation system closely followed the national policies and measures; moreover, it realized the thoughts of building a "double-carbon" supply chain performance evaluation index system from different levels of supply-chain benefits, and it built a performance evaluation system for the energy supply chain from four dimensions: energy supply, energy production and processing, energy transmission and distribution and energy consumption, which was an innovative idea for establishing a performance evaluation system for the energy supply chain [9,10].

## 2. Literature Review

In many countries, research on the energy supply chain has been more inclined to use various model methods to assess and prevent the risks of the energy supply chain and to study the optimal operation of the specific type of the energy supply chain (such as a coal-fired power supply chain, bioenergy supply chain, etc.). Some cases in point are as follows: Gernaat et al., used climate and integrated assessment models to estimate this effect on key renewables [11]; Iqbal et al., modeled by using the sigmoid function [12]; Carolin Monsberger et al., applied a mixed-integer linear optimization model [13]; Murele et al., derived a supply chain model and the governing equations by using general algebraic modeling system (GAMS) software [14]; Barragán-Escandón et al., utilized the long-range energy alternative planning (LEAP) model [15]. Beyond these, scholars have also used other different types of models to deal with the algorithm problems related to different types of energy supply chains.

Additionally, scholars' research on energy supply chains can generally be divided into two categories: some discuss a specific type of energy supply chain, such as the bioenergy supply chain and coal-energy supply chain; others research the risk influencing factors and the mechanism of the energy supply chain in a broad sense and analyze the opportunities and challenges that are faced by energy industries, such as the "double-carbon" goals. However, there are less related research results on the construction of a performance evaluation system for the energy supply chain.

For example, Meng et al., put forward the main problems that exist in the energy supply chain of the iron and steel complex from four aspects: raw material supply, node connection, management and environment and provision of the optimization strategy [16]. Mohamed Rimsan et al., proposed that the Ethereum blockchain platform should be combined with existing traditional infrastructure and a unique identity and intelligent contract should be used to track and investigate energy-supply-chain activities to ensure the availability of anti-risk data on the energy supply chain [17]. Emenike et al., cited the scientific work on the elasticity of biomass, water, the power system, natural gas and the energy supply chain, and they studied the elasticity of the energy supply chain based on optimization so as to continuously realize the elasticity of the energy supply chain in activities, such as production, storage and transportation [18]. Xin Zhang took "the belt and road initiative" as the background, analyzed the network structure of the energy supply chain under the background of "the belt and road initiative" by using the small-world network model and screened and analyzed the risk factors that affected the energy supply chain [19]. Yang Yang et al., established the elasticity measurement model of a regional energy supply chain by using Bayesian posterior probability and they made an empirical study on it with the relevant data on Beijing from the perspective of regional resource limitation [20]. Guo Yu et al., simulated the evolution of the energy system from the angle of the "Chinese energy prospect model", concluded that the goals of "double-carbon" should require large-scale non-mineral energy and clean and low-carbon fossil energy and summarized the existing problems, development requirements and measures in the Chinese energy industry [21].

The opportunities and challenges faced by energy industries against the background of "double-carbon" have been intensely studied, with previous research findings providing ideas and a theoretical basis for this paper so that we could carry out a broader and more effective scientific analysis. However, there is still a gap in terms of the performance evaluation of the energy supply chain based on "double-carbon" goals, as most studies focused more on the low-carbon transformation of the whole energy supply chain [21,22]. The literature has not fully considered the characteristics of the energy supply chain, studied the performance evaluation of the energy supply chain in detail or formed a comprehensive, systematic and scientific performance evaluation system that is applicable to the energy supply chain.

In light of the above background and the necessity of the "double-carbon" transformation of the energy industry, this paper analyzes each node in the energy supply chain, combined with the development trend of the rigid requirements of "double-carbon," as well as the main index factors and related problems affecting the overall performance of the energy supply chain at the levels of energy development, energy production and processing, energy transmission and distribution and energy consumption. In order to achieve "double-carbon" goals, we take the energy supply chain as the research object, build a performance evaluation index system of an energy supply chain that complied with the requirements of the latest national policies, obtain evaluation data through the development of Chinese energy industry in each year, analyze the level of the Chinese energy industry and provide relevant suggestions for adjusting Chinese energy supply and demand. We discuss the problems of insufficient energy supply and demand and realize the green transformation of the energy industry under the "double-carbon" goals, which could be more scientific and efficient [23–25].

Against the backdrop of "double-carbon" goals, this study aimed to build a more detailed, targeted and comprehensive performance evaluation index system for the energy supply chain, which fully considers the main indicators in the energy supply chain. In addition, based on traditional data envelopment analysis, a three-stage DEA model (including the DEA-BCC model [26], the DEA-CCR model [27], and SFA regression) was adopted, eliminating the influence of environmental factors, thus, allowing us to compare the performance of the energy supply chain [28,29].

## 3. Materials and Methods

Chinese President Xi introduced the concepts of carbon peaking and carbon neutrality (double-carbon) at the 75th general debate of the United Nations General Assembly [30]. "Carbon peaking" refers to the peaking of carbon dioxide emissions at a specific point in time and with a downward trend. "Carbon neutrality" refers to the total amount of greenhouse gas emissions directly or indirectly generated by enterprises, collectives or individuals in a certain period of time, offset through afforestation, energy conservation and emission reduction, so as to achieve the goal of "zero emissions" [2], as shown in Figures 1 and 2.

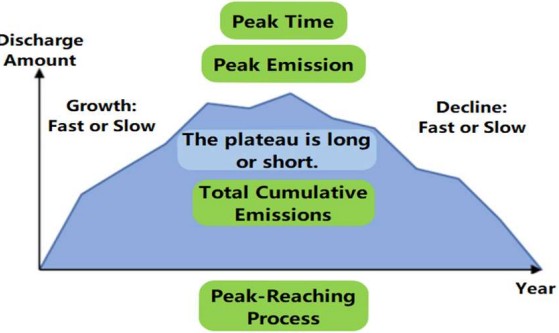

**Figure 1.** Simple diagram of "peak carbon dioxide emissions" process.

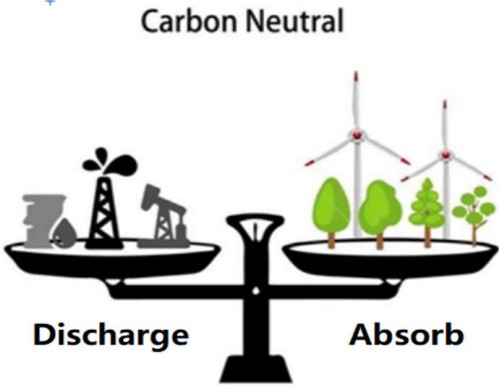

**Figure 2.** Simple schematic diagram of "carbon neutralization".

In the traditional energy supply chain, the upstream of the traditional energy supply chain generally includes coal production enterprises and some metal- and material-processing enterprises. In the middle reaches, they are transported to power plants for power generation by waterways, railways, etc., then to the downstream cable transmission and distribution and finally to residential and industrial power consumption [30]. The supply chain structure is shown in Figure 3.

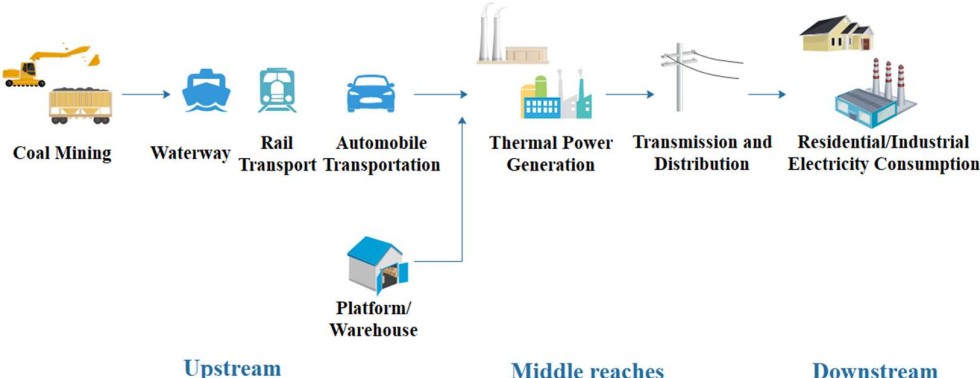

**Figure 3.** Traditional energy supply chain.

Compared with the traditional energy supply chain, the key difference of the new energy supply chain lies in the emphasis on the development and utilization of clean energy such as photovoltaic and wind power, and the continuous improvement and innovation of electric energy storage technology [31]. Its basic architecture is shown in Figure 4.

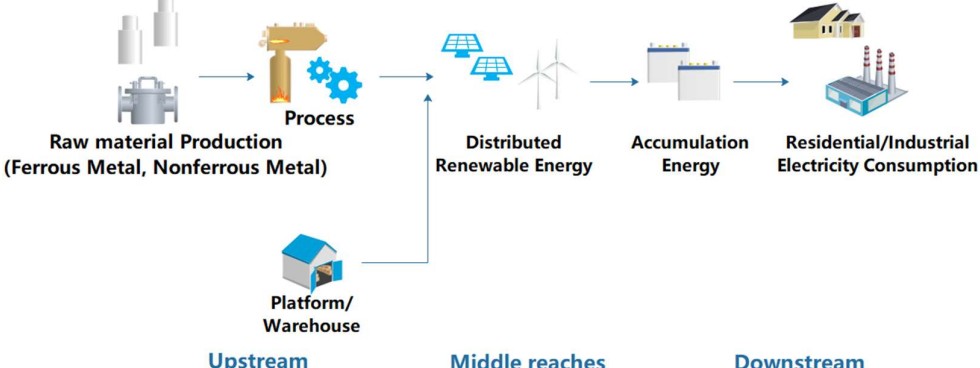

**Figure 4.** New energy supply chain.

*3.1. Theoretical Model*

3.1.1. Analytic Hierarchy Process (AHP)

In this paper, the analytic hierarchy process (AHP) was used to establish a model to solve the weight of energy supply chain performance evaluation indicators under the "double-carbon" goals. Its implementation steps include:

Establishing a Hierarchical Structure Model

The overall hierarchical structure system is divided into three levels: the highest level (target level), the intermediate level (standard level) and the element level (index level).

Constructing a Judgment Matrix

Each element in the judgment matrix indicates the relative importance of the two corresponding indexes to the upper indexes. Comparing the importance of the two factors and *j* which is relative to the total objective, the element $a_{ij}$ of the pairwise comparison matrix is given on a 1–9 scale devised by Satty (1980), shown in Table 1 [29].

**Table 1.** Paired comparison matrix scale table.

| Scale | Meaning |
|:---:|:---:|
| 1 | The *i* factor has the same influence as the *j* factor. |
| 3 | The *i* factor has a slightly stronger influence than the *j* factor. |
| 5 | The *i* factor has a stronger influence than the *j* factor. |
| 7 | The *i* factor has a obviously stronger influence than the *j* factor. |
| 9 | The influence of the *i* factor is extremely stronger than the *j* factor. |
| 2, 4, 6, 8 | The median value of the above adjacent judgment |
| reciprocal | The judgment of comparing factor *i* with *j* is $a_{ij}$, and the judgment $a_{ji} = 1/a_{ij}$ of comparing factor *i* with *j*. |

Hierarchical Single Sorting and Its Consistency Test

Let the maximum eigenvalue of judgment matrix $A$ be $\lambda_{\max}$ and the eigenvector be $W$. After normalization, it is the relative importance ranking of each factor at the same level relative to the factors at the previous level. In this paper, the square root method is adopted to sort the hierarchical list and its calculation steps are as follows [29]:

Calculate the product $M_i$ of elements in each row of judgment matrix $A$:

$$M_i = \prod_{i=1}^{n} (i = 1, 2, \ldots, n) \tag{1}$$

Calculate the nth root of $M_i$:

$$\overline{W_i} = \sqrt[n]{M_i}(i = 1, 2, \ldots, n) \tag{2}$$

$$W_i = w_i / \sum_{i=1}^{n} \overline{w_i}(i = 1, 2, \ldots, n); W = (w_1, w_2, \ldots, w_n)^T \tag{3}$$

After the vector is normalized, the characteristic vector $W$ is obtained:

The decision matrix must pass the consistency test: only a decision matrix that meets the consistency test has practical significance. The calculation steps include:

Calculate the maximum eigenvalue $\lambda_{\max}$ of the judgment matrix $A$:

$$\lambda_{\max} = \sum_{i=1}^{n} \frac{(Aw)_i}{nw_i} \tag{4}$$

Calculate the consistency index $CI$:

$$CI = \frac{\lambda_{\max} - n}{n - 1} \tag{5}$$

Calculate random consistency ratio ($CR$):

$$CR = \frac{CI}{RI} \tag{6}$$

This is related to the order $n$ of the average random consistency test value index ($RI$) of the judgment matrix and the $RI$ comparison table given by Santy [29] is as follows in Table 2.

**Table 2.** Average random consistency test value index (confidence level is 90%).

| $n$ | 1 | 2 | 3 | 4 | 5 | 6 | 7 | 8 |
|-----|---|---|-----|------|------|------|------|------|
| RI | 0 | 0 | 0.52 | 0.89 | 1.12 | 1.26 | 1.36 | 1.41 |

Only when $CR < 0.1$ is the consistency of the judgment matrix good or its inconsistency acceptable; otherwise, it is necessary to adjust the judgment matrix until satisfactory consistency is obtained.

General Ranking and Its Consistency Test

Generally, an AHP analysis contains two levels of model structure. Therefore, it is necessary to synthesize the obtained single criteria weights from bottom to top to form a total hierarchical ranking. The method and calculation steps are consistent with the above hierarchical single ranking.

3.1.2. Three-Stage DEA Analysis

The three-stage DEA method is divided into the first-stage DEA model, the second-stage SFA (stochastic frontier analysis) model and the third-stage DEA model. It can adjust the influence of environmental variables and random interference items through the second-stage SFA model and make up for the deficiency in the traditional DEA model that does not consider environmental variables and random errors.

The First Stage: Measuring the Initial Efficiency Value Using DEA-BCC Model and DEA-CCR Model

The DEA-BCC model (extended by Banker, Charnes and Cooper) [26] and DEA-CCR model (proposed by Charnes, Cooper and Rhodes) [27] are used to calculate the initial input–output efficiency and the BCC model is used to analyze the comprehensive technical efficiency (TE). Then, the scale efficiency (SE) is obtained by comparing the results calculated by the CCR model with comprehensive technical efficiency; the pure technical efficiency (PTE) is the ratio of technical efficiency to scale efficiency. The lower the comprehensive efficiency, the lower the efficiency of resource allocation and vice versa. The specific models are as follows [32]:

$DEA - BC^2$ Model:

$$\max_{\Phi}, \lambda \Phi, \tag{7}$$

s.t.

$$-\Phi q_i + Q\lambda \geq 0 \tag{8}$$

$$x_i - X\lambda \geq 0 \tag{9}$$

$$\sum \lambda = 1 \tag{10}$$

$$\lambda \geq 0 \tag{11}$$

$DEA - C^2R$ Model:

$$\max_{\Phi}, \lambda\Phi, \tag{12}$$

s.t.

$$-\Phi q_i + Q\lambda \geq 0 \tag{13}$$

$$x_i - X\lambda \geq 0 \tag{14}$$

$$\lambda \geq 0 \tag{15}$$

Among them, the subvector $X$ contains observed input values $x_i$ [26], $\Phi$ is the efficiency evaluation value, $q$ is the explained unit variable, $\lambda$ is the weight and $Q$ is the explanatory variable. $\Phi$ is between 0 and 1; if $\Phi = 1$, the efficiency of resource allocation is the highest; if $\Phi = 0$, the efficiency of resource allocation is the lowest.

The Second Stage: SFA Regression Analysis Eliminates the Influence of Environmental Noise Interference

In the first stage, relaxation variables are obtained, which can reflect the influence of environmental factors, low management efficiency, statistical noise and other factors on the initial investment. In this paper, the SFA regression method is used to express relaxation variables as three effects and the formula is as follows [32,33]:

$$S_{ni} = f(Z_i; \beta_n) + v_{ni} + \mu_{ni}; i = 1, 2, \ldots, I; n = 1, 2, \ldots, N \tag{16}$$

where $S_{ni}$ is the slack of the input item $n$ of the $i$th decision-making unit, $Z_i$ is the environmental variable, $\beta_n$ is the coefficient of the environmental variable, $v_{ni} + \mu_{ni}$ is the mixed error term, $v_{ni}$ represents random interference and $\mu_{ni}$ represents management inefficiency [34].

Then, according to the formula of Dengyue Luo [35,36], the inefficient items $\mu$ are separated and managed.

The separation formula for conditional estimation of management inefficiency $E(\mu|\varepsilon)$ is:

$$E(\mu|\varepsilon) = \sigma_* \left[ \frac{\phi(\lambda\frac{\varepsilon}{\sigma})}{\Phi(\frac{\lambda\varepsilon}{\sigma})} + \frac{\lambda\varepsilon}{\sigma} \right] \tag{17}$$

in which, $\sigma$ represents variance and $\varepsilon$ represents comprehensive error [36], $\sigma_* = \frac{\sigma_\mu \sigma_v}{\sigma}$, $\sigma = \sqrt{\sigma_\mu^2 + \sigma_v^2}$, $\lambda = \sigma_\mu / \sigma_v$.

Then, the random error term $v_{ni} + \mu_{ni}$ is:

$$E[v_{ni}|v_{ni} + \mu_{ni}] = S_{ni} - f(Z_i; \beta_n) - E[\mu_{ni}|v_{ni} + \mu_{ni}] \tag{18}$$

The Third Stage: Calculate the Adjusted DEA Model Efficiency Value

After adding the original data to the random error term obtained in the second stage, the adjusted input–output quantity $(X_{ni}^A)$ can be obtained and the adjustment formula is as follows [37]:

$$X_{ni}^A = X_{ni} + [\max(f(Z_i; \hat{\beta}_n)) - f(Z_i; \hat{\beta}_n)] + [\max(v_{ni}) - v_{ni}] \; i = 1, 2, \cdots, I; n = 1, 2, \cdots, N \tag{19}$$

That is, the adjusted input value is equal to the original input value $(X_{ni})$ plus the adjusted value of environmental variables plus the adjusted value of random interference, then the adjusted data are obtained. As with the steps of the first stage, we should calculate the adjusted efficiency values and compare them with those before adjustment.

*3.2. Comprehensive Performance Evaluation Model Based on AHP and Three-Stage DEA*

In this paper, based on AHP and three-stage DEA, the comprehensive performance evaluation model of the energy supply chain under the "double-carbon" goals has basic operation steps as follows [37–40]:

- Using the AHP method, calculate the weight $W_i$ of each first-level index relative to the total target;
- Classify the secondary indicators and construct the evaluation set of decision-making units of each indicator;
- Considering the influence of two environmental variables, which are the GDP growth rate and the total population growth rate at the end of the year, this paper uses a three-stage DEA model to calculate the technical efficiency ($c_{ij}$) under the fixed return on scale model (namely the $C^2R$ model, hereinafter abbreviated CRS) and the technical efficiency ($v_{ij}$) and scale efficiency ($s_{ij}$) under the variable return on scale model (i.e., the $BC^2$ model, hereinafter abbreviated VRS) of each first-level index, respectively, indicating that the jth energy supply chain has an impact on the ith first-level index.
- Using the weight $W_i$ obtained by the above calculation and the efficiency evaluation values, we calculate and compare the comprehensive efficiency values $C_j$, $V_j$ and $S_j$ of the jth energy supply chain ($j = 1, \ldots, 10$, where $j = 1$ means the Chinese energy supply chain in 2010 and so on). The calculation formulas are as follows:

$$C_j = \sum_{i=1}^{m} W_i c_{ij}, j = 1, 2, \ldots, n \tag{20}$$

$$V_j = \sum_{i=1}^{m} W_i v_{ij}, j = 1, 2, \ldots, n \tag{21}$$

$$S_j = \sum_{i=1}^{m} W_i s_{ij}, j = 1, 2, \ldots, n \tag{22}$$

## 4. Construction of Performance Evaluation Index System of Energy Supply Chain under "Double-Carbon" Goals

### 4.1. Constructing an Index System Based on Energy Supply Chain Architecture

By consulting a large number of related studies, starting with the main economic production activities at each node, we extracted the factors that might affect the overall operation performance at each node of the energy supply chain. At the same time, we considered the operability, completeness and accuracy of data collection in the later stage and, being able to clearly divide the input from output indicators, we selected and screened the indicators based on the four-tier structure of the energy supply chain [41].

#### 4.1.1. Energy Supply

The analysis of the energy supply mainly includes the raw material supply and the dependence on imported energy. The purpose is to estimate the resources and economic and technological potential of the energy supply and to evaluate the balance between supply and demand. From the perspective of raw material supply, it will involve two aspects: raw material supply quantity and raw material types. Therefore, "total primary energy production (10,000 tons of standard coal)" and "low-carbon raw material rate" are selected. Considering the dependence on imported energy supply, the quantitative index "energy self-sufficiency rate" is selected [42–44].

#### 4.1.2. Energy Production and Treatment

In this paper, for the analysis of energy production and treatment, the indicators are selected from three aspects: production and treatment efficiency, pollutant discharge and residual energy recovery after production. After consulting the literature and considering all feasible indicators comprehensively, we selected "energy processing conversion efficiency," "carbon emission rate per unit output," and "energy recovery rate," respectively, to reflect them [45–49].

### 4.1.3. Energy Transmission and Distribution

In this paper, the quality of the energy supply chain in the stage of transmission and distribution is reflected by selecting indicators from two main angles, energy dispatch and transmission efficiency. For the efficiency of energy dispatching, the index "dispatching extensiveness" is selected and quantified. Through the analysis of the relevant data indicators released by the National Energy Statistics Bureau, considering its meaning and collectability, the index data of "transmission line loop length" finally reflect the scope of dispatching construction. As for the transmission quality of energy resources, it is mainly delivered to users in the form of electric power or heat energy for use and most of them are electric power [50]. For the limitation of data, we selected the "(electric power) transmission arrival rate" index to reflect it [51]. At the same time, the "energy storage rate" can reflect the effect of energy reserves in the energy supply chain.

### 4.1.4. Energy Consumption

The analysis of energy consumption mainly focuses on consumers and the selected indicators come from three aspects: consumption level, supply and demand balance and low-carbon consumption. The variables of "per capita energy consumption," "demand satisfaction rate," and "proportion of clean energy consumption" are selected to reflect the energy consumption in the energy supply chain [17].

### 4.2. Performance Evaluation Index System of Energy Supply Chain under the Goal of "Double-Carbon"

Based on a detailed analysis of the above subnodes, the energy supply, the energy production and processing, the energy transmission and distribution and the energy consumption are taken as the first-level indicators and second-level indicators, which mainly affect the overall performance of the energy supply chain and are gradually selected from the basic structure of the energy supply chain, forming the performance evaluation index system of the energy supply chain under the "double-carbon" goals, as shown in Figure 5.

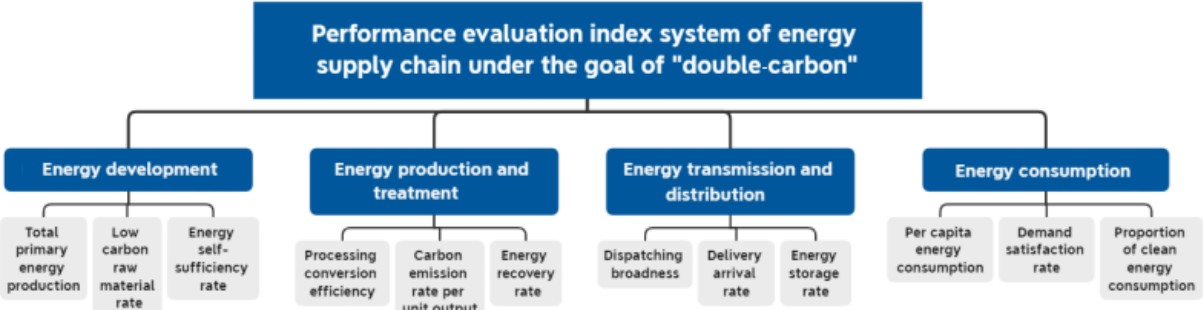

**Figure 5.** Performance evaluation index system of energy supply chain under "double- carbon" goals.

## 5. Relative Performance Evaluation of Chinese Energy Supply Chain Based on Three-Stage DEA

### 5.1. Data Acquisition and Description

Based on the evaluation index system established above, we collected and collated the relevant index data from the Chinese energy industry from 2010 to 2019 and comprehensively evaluated the performance of the Chinese energy supply chain in recent years.

All the initial data in Table 3 were selected from the annual statistical data given in the *China Energy Statistical Yearbook*. A small amount of missing data was supplemented by website searches and the ratio data were calculated by a comparison with the data in the *China Energy Statistical Yearbook*.

**Table 3.** List of quantitative data.

| Year | Energy Supply | | | Energy Production and Treatment | | | Energy Transmission and Distribution | | | Energy Consumption | | |
|---|---|---|---|---|---|---|---|---|---|---|---|---|
| | 1 | 2 | 3 | 4 | 5 | 6 | 7 | 8 | 9 | 10 | 11 | 12 |
| 2010 | 312,125 | 10.40 | 88.10 | 72.50 | 2.46 | 2.87 | 438,732.00 | 99.82 | 0.0013 | 2696 | 86.55 | 11.75 |
| 2011 | 340,178 | 9.60 | 86.30 | 72.20 | 2.43 | 2.97 | 474,937.00 | 99.77 | 0.0007 | 2880 | 87.89 | 13.00 |
| 2012 | 351,041 | 11.20 | 85.60 | 72.70 | 2.43 | 3.20 | 505,640.00 | 99.56 | 0.0017 | 2977 | 87.29 | 14.50 |
| 2013 | 358,784 | 11.80 | 84.60 | 73.00 | 2.45 | 3.72 | 543,896.00 | 98.36 | 0.0019 | 3071 | 86.06 | 15.50 |
| 2014 | 362,212 | 13.30 | 84.00 | 73.10 | 2.44 | 4.03 | 577,605.00 | 97.32 | 0.0025 | 3140 | 84.48 | 17.00 |
| 2015 | 362,193 | 14.50 | 84.20 | 73.40 | 2.46 | 4.01 | 607,643.00 | 96.95 | 0.0035 | 3166 | 83.27 | 18.00 |
| 2016 | 345,954 | 16.80 | 79.40 | 73.50 | 2.57 | 4.41 | 645,609.00 | 96.50 | 0.0056 | 3202 | 78.38 | 19.50 |
| 2017 | 358,867 | 17.40 | 80.00 | 73.00 | 2.52 | 4.43 | 685,567.00 | 96.35 | 0.0082 | 3288 | 78.76 | 20.80 |
| 2018 | 378,859 | 18.00 | 79.00 | 72.80 | 2.47 | 5.43 | 724,788.00 | 96.85 | 0.0176 | 3388 | 79.89 | 22.10 |
| 2019 | 397,317 | 18.40 | 81.70 | 73.30 | 2.40 | 5.38 | 759,465.00 | 97.09 | 0.0207 | 3488 | 81.44 | 23.40 |

Notes: 1—Total primary energy production (10,000 tons of standard coal). 2—Low carbon raw material rate. 3—Energy self-sufficiency rate. 4—Energy processing conversion efficiency. 5—Carbon emission rate per unit output. 6—Energy recovery rate. 7—Dispatching broadness (transmission line loop length in km). 8—Timely delivery rate. 9—Energy storage rate (MV/10,000 tons of standard coal). 10—Per capita energy consumption (kg standard coal). 11—Demand satisfaction rate. 12—Proportion of clean energy consumption (clean energy is nonpollutant energy, which mainly includes nuclear energy and "renewable energy").

### 5.2. Determination of First-Level Index Weight

By comprehensively considering the weight setting and ranking conclusions of relevant studies [17,20,21,24,30,36,37] and according to the basic principles and calculation steps of the above analytic hierarchy process, the influence degree of each node in the energy supply chain on the overall operation performance of the supply chain was roughly defined, which we used to determine the relative weight of each indicator for the performance evaluation of the energy supply chain under the "double-carbon" goals.

Then, the judgment matrix $A$ was constructed for energy supply, energy production and processing, energy transmission and distribution and energy consumption under the performance evaluation index system of the energy supply chain under the "double-carbon" goals at the target level as follows:

$$A = \begin{bmatrix} 1 & 1/3 & 3 & 5 \\ 3 & 1 & 5 & 6 \\ 1/3 & 1/5 & 1 & 3 \\ 1/5 & 1/6 & 1/3 & 1 \end{bmatrix}$$

We used the square root method to calculate the judgment matrix and obtained the relative weight $W_i$ of the first-level index as follows:

$$W_i = (0.269, 0.553, 0.120, 0.058) \tag{23}$$

The weight of the obtained first-level index was checked for consistency and $\lambda_{\max} = 4.1501$ was calculated.

$$CI = \frac{\lambda_{\max} - n}{n - 1} = \frac{4.1501 - 4}{4 - 1} = 0.0500 \tag{24}$$

When $n = 4$, with a confidence level of 90%, the look-up table shows a critical value of $RI = 0.89$, then

$$CR = \frac{CI}{RI} = \frac{0.0500}{0.89} = 0.0562 < 0.1 \tag{25}$$

Therefore, the judgment matrix has good consistency and the calculated $W_i$ is acceptable:

$$W_i = (0.2685, 0.5531, 0.1201, 0.0583) \tag{26}$$

According to the obtained weight vector, the index of energy development and processing had the highest weight, 0.5531, indicating that in the operation of the energy supply

chain, the energy development and processing link mainly affected the overall performance of energy supply chain and was the critical link to realize the transformation of the energy supply chain toward "double-carbon" goals, followed by energy supply.

### 5.3. Relative Performance Evaluation of Chinese Energy Supply Chain
#### 5.3.1. Establishment of Input–Output Index Evaluation Set

According to the evaluation objective of "the fewer resources invested, the greater output," taking into account two environmental variables of GDP growth rate and population growth rate at the end of the year, the evaluation set of input–output indicators of each decision-making unit ($DMU$) is constructed in Table 4.

**Table 4.** Classification table of input–output index evaluation under each first-level index.

| Primary Index | Input/Output Index | Secondary Index |
|---|---|---|
| Energy supply (0.2685) | Input index | Total primary energy production |
| | Output index | Low carbon raw material rate Energy self-sufficiency rate |
| Energy production and treatment (0.5531) | Input index | Carbon emission rate per unit output |
| | Output index | Energy processing conversion efficiency Energy recovery rate |
| Energy transmission and distribution (0.1201) | Input index | Dispatching broadness |
| | Output index | Delivery arrival rate Energy storage rate |
| Energy consumption (0.0583) | Input index | Per capita energy consumption |
| | Output index | Demand satisfaction rate Proportion of clean energy consumption |
| environmental index | | GDP growth rate Total population growth at the end of the year |

#### 5.3.2. Relative Performance Evaluation of Chinese Energy Supply Chain

We selected the Chinese energy supply chain from 2010 to 2019 as the decision-making unit ($DMU_i, i = 1, 2, \ldots, 10$) to evaluate and establish a three-stage DEA model.

The first stage: calculate the values of comprehensive technical efficiency, pure technical efficiency and scale efficiency. For example, according to the input-output classification table, the original data of the secondary indicators under the energy production processing plus environmental variables were sorted, as in Table 5. Similarly, other original data could be obtained. After classifying all of the data from 2010 to 2019 under each first-level index, we stored all of them in Deap2.1 software to get the relative efficiency value of each first-level index and calculated the comprehensive relative efficiency value according to the above Equations (20)–(22), as shown in Table 6.

In Table 6, the overall situation of Chinese energy supply chain efficiency from 2010 to 2019 is as follows: the technical efficiency is 0.953, the pure technical efficiency is 0.993 and the scale efficiency is 0.959. The technical efficiency, pure technical efficiency and scale efficiency are all very close to the frontier standards of efficiency. In contrast, the technical efficiency shows great room for improvement. Generally speaking, in the past 10 years, considering the "double-carbon" goals, the Chinese energy supply chain has exhibited a high level of performance and good overall operations.

The second stage: classify the relaxation variables and environmental variables of carbon emission rate per unit in the first stage in Table 7 and adjust the data using the Stochastic Frontier Model (SFA). Use Frontier 4.1 to import the original data into DTA, set the INS script and run it to get the results of each input. The main results are summarized in Table 8.

**Table 5.** The original data of the secondary indicators under the energy production processing plus environmental variables.

| Year | Output Index | | Input Index | Environmental Variables | |
| --- | --- | --- | --- | --- | --- |
| | Energy Processing Conversion Efficiency | Energy Recovery Rate | Carbon Emission Rate per Unit Output | GDP Growth Rate | Total Population Growth at the End of the Year |
| 2010 | 72.50 | 2.87 | 2.46 | 0.1825 | 0.0048 |
| 2011 | 72.20 | 2.97 | 2.43 | 0.1840 | 0.0048 |
| 2012 | 72.70 | 3.20 | 2.43 | 0.1038 | 0.0050 |
| 2013 | 73.00 | 3.72 | 2.45 | 0.1010 | 0.0049 |
| 2014 | 73.10 | 4.03 | 2.44 | 0.0853 | 0.0052 |
| 2015 | 73.40 | 4.01 | 2.46 | 0.0704 | 0.0050 |
| 2016 | 73.50 | 4.41 | 2.57 | 0.0835 | 0.0059 |
| 2017 | 73.00 | 4.43 | 2.52 | 0.1147 | 0.0053 |
| 2018 | 72.80 | 5.43 | 2.47 | 0.1049 | 0.0038 |
| 2019 | 73.30 | 5.38 | 2.40 | 0.0731 | 0.0033 |

**Table 6.** Comprehensive efficiency value of energy supply chain in the first stage.

| DMU | $C_j$ | $V_j$ | $S_j$ |
| --- | --- | --- | --- |
| 2010 | 0.981 | 0.993 | 0.987 |
| 2011 | 0.947 | 0.986 | 0.961 |
| 2012 | 0.944 | 0.990 | 0.954 |
| 2013 | 0.930 | 0.988 | 0.941 |
| 2014 | 0.935 | 0.989 | 0.945 |
| 2015 | 0.940 | 0.995 | 0.945 |
| 2016 | 0.936 | 0.997 | 0.938 |
| 2017 | 0.945 | 0.994 | 0.950 |
| 2018 | 0.980 | 0.999 | 0.981 |
| 2019 | 0.988 | 1.000 | 0.988 |
| mean | 0.953 | 0.993 | 0.959 |

**Table 7.** SFA regression data of carbon emission rate per unit in the second stage.

| DMU | Number | Time | Investment in Fixed Assets | GDP Growth Rate | Total Population Growth at the End of the Year |
| --- | --- | --- | --- | --- | --- |
| 2010 | 1 | 1 | −0.0038 | 18.2490 | 0.4803 |
| 2011 | 2 | 1 | −0.0040 | 18.3978 | 0.4803 |
| 2012 | 3 | 1 | −0.0028 | 10.3783 | 0.4965 |
| 2013 | 4 | 1 | 0.0001 | 10.0975 | 0.4933 |
| 2014 | 5 | 1 | −0.0029 | 8.5334 | 0.5218 |
| 2015 | 6 | 1 | −0.0049 | 7.0382 | 0.4971 |
| 2016 | 7 | 1 | −0.0020 | 8.3525 | 0.5885 |
| 2017 | 8 | 1 | −0.0009 | 11.4740 | 0.5330 |
| 2018 | 9 | 1 | 0.0005 | 10.4857 | 0.3813 |
| 2019 | 10 | 1 | −0.0030 | 7.3138 | 0.3347 |

The third stage: calculate the adjusted comprehensive efficiency values. According to the adjusted data, the same as the traditional DEA model calculation method and software operation in the first stage, the results are as follows in Tables 9 and 10.

**Table 8.** Regression estimation table of environmental variables to input relaxation variables.

| | Total Primary Energy Production | Carbon Emission Rate Per Unit Output | Scheduling Reasonable Rate | Per Capita Energy Consumption |
|---|---|---|---|---|
| coefficient | 25,920.2670 | −0.0001 | 152,152.2900 | 11.3765 |
| GDP growth rate | −607.7949 | −0.0001 | 7079.2817 | −0.2083 |
| Total population growth at the end of the year | −30,857.7520 | −0.0022 | −621,362.4300 | −15.6923 |
| sigma-squared | 365,948,740.0000 | 0.0000 | 344,854,240.0000 | 217.1391 |
| gamma | 1.0000 | 0.0500 | 0.0495 | 1.0000 |

**Table 9.** Adjusted comprehensive efficiency value in the third stage.

| DMU | $C_j$ | $V_j$ | $S_j$ |
|---|---|---|---|
| 2010 | 0.981 | 0.993 | 0.987 |
| 2011 | 0.950 | 0.986 | 0.964 |
| 2012 | 0.949 | 0.990 | 0.958 |
| 2013 | 0.938 | 0.988 | 0.948 |
| 2014 | 0.939 | 0.989 | 0.949 |
| 2015 | 0.942 | 0.994 | 0.947 |
| 2016 | 0.953 | 0.998 | 0.956 |
| 2017 | 0.955 | 0.994 | 0.961 |
| 2018 | 0.989 | 1.000 | 0.989 |
| 2019 | 0.990 | 1.000 | 0.990 |
| mean | 0.958 | 0.993 | 0.965 |

**Table 10.** Relative efficiency value for each primary index in the third stage.

| DMU | Energy Supply (0.2685) | | | Energy Production and Treatment (0.5513) | | | Energy Transmission and Distribution (0.1021) | | | Energy Consumption (0.0583) | | |
|---|---|---|---|---|---|---|---|---|---|---|---|---|
| | $c_{ij}$ | $v_{ij}$ | $s_{ij}$ | $c_{ij}$ | $v_{ij}$ | $s_{ij}$ | $c_{ij}$ | $v_{ij}$ | $s_{ij}$ | $c_{ij}$ | $v_{ij}$ | $s_{ij}$ |
| 2010 | 1 | 1 | 1 | 0.965 | 0.988 | 0.977 | 1 | 1 | 1 | 1 | 1 | 1 |
| 2011 | 0.901 | 0.98 | 0.92 | 0.973 | 0.984 | 0.988 | 0.941 | 0.999 | 0.942 | 0.979 | 1 | 0.979 |
| 2012 | 0.904 | 0.981 | 0.921 | 0.98 | 0.991 | 0.988 | 0.892 | 0.997 | 0.894 | 0.982 | 1 | 0.982 |
| 2013 | 0.892 | 0.975 | 0.914 | 0.98 | 0.995 | 0.985 | 0.832 | 0.985 | 0.844 | 0.969 | 0.994 | 0.975 |
| 2014 | 0.914 | 0.982 | 0.931 | 0.981 | 0.996 | 0.985 | 0.784 | 0.975 | 0.804 | 0.974 | 0.988 | 0.985 |
| 2015 | 0.947 | 0.994 | 0.953 | 0.977 | 1 | 0.977 | 0.75 | 0.971 | 0.772 | 0.982 | 0.987 | 0.995 |
| 2016 | 1 | 1 | 1 | 0.94 | 1 | 0.94 | 0.9 | 0.98 | 0.918 | 0.975 | 1 | 0.975 |
| 2017 | 0.999 | 1 | 0.999 | 0.952 | 0.994 | 0.958 | 0.858 | 0.979 | 0.876 | 0.982 | 1 | 0.982 |
| 2018 | 0.993 | 1 | 0.993 | 0.985 | 1 | 0.985 | 1 | 1 | 1 | 0.99 | 1 | 0.99 |
| 2019 | 0.977 | 1 | 0.977 | 1 | 1 | 1 | 0.964 | 1 | 0.964 | 1 | 1 | 1 |
| mean | 0.953 | 0.991 | 0.961 | 0.973 | 0.995 | 0.978 | 0.892 | 0.989 | 0.901 | 0.983 | 0.997 | 0.986 |

Combine the DEA efficiency results of the first and third stages in one line chart, as shown in the following figures.

It can be seen from Figures 6–8 that, after adjusting and eliminating the influence of environmental variables and random factors, the overall situation in the Chinese energy supply chain in 2010–2019 had a technical efficiency value of 0.958, a pure technical efficiency value of 0.993 and a scale efficiency value of 0.965. Compared with the previous adjustment, the technical efficiency and scale efficiency slightly increased, while the pure technical efficiency remained unchanged. The difference between the overall situation before and after the adjustment was not large, which preliminarily indicates that the two environmental variables, GDP growth rate and population growth rate at the end of the year, have little impact on the overall efficiency of the Chinese energy supply chain.

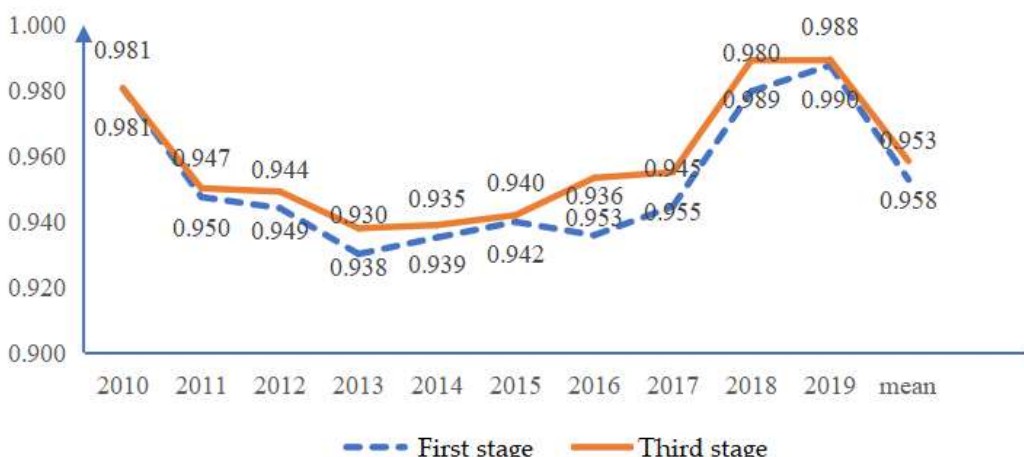

**Figure 6.** Line chart before and after adjustment of comprehensive technical efficiency.

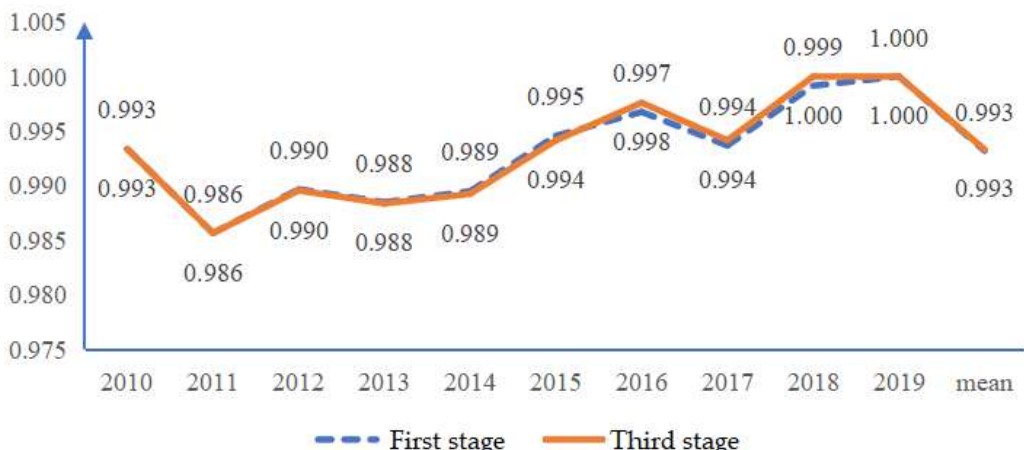

**Figure 7.** Line chart before and after adjustment of comprehensive pure technical efficiency.

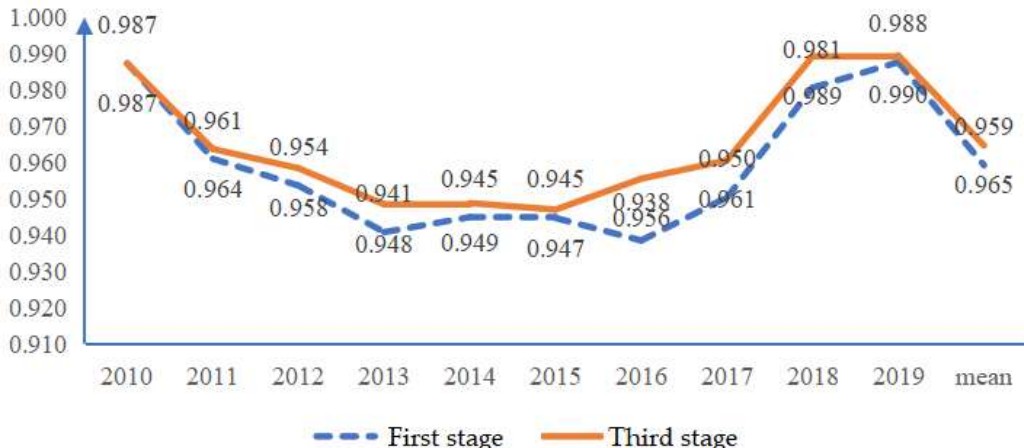

**Figure 8.** Line chart before and after adjustment of comprehensive scale efficiency.

Technical efficiency and scale efficiency increased the most in 2013, 2018 and 2019, which indicated that the GDP growth rate and the population growth rate at the end of these three years were higher than in other years and had a significantly positive impact on the performance of its energy supply chain.

*5.4. Comparative Analysis of Traditional Energy Supply Chain and New Energy Supply Chain*

The new Chinese energy industry officially entered a high-speed development stage in 2011. We define 2010–2014 as the era of the traditional energy supply chain and 2015–2019 as the era of the new energy supply chain.

According to the relative efficiency values of each node in the Chinese energy supply chain, generally speaking, the technical efficiency, pure technical efficiency and scale efficiency of each node in the era of the new energy supply chain are at the forefront of efficiency in most years and their efficiency is generally higher than that of the traditional energy supply chain, especially on the energy supply side. Since 2015, the operational performance of the Chinese energy supply chain has improved rapidly, reaching the forefront of efficiency in 2019. The essential difference between the new energy supply chain and the traditional energy supply chain is the raw materials. The new energy supply chain focuses on the conversion and utilization of photovoltaic, hydro-energy, wind energy, etc. The use of clean raw materials can greatly increase the proportion of low-carbon raw materials in the energy supply and reduce the dependence on imported primary energy sources, such as coal and oil, which promotes the transformation to low-carbon options in the Chinese energy industry [48–51].

For energy production and treatment, the processing and conversion technology in the new era has been greatly improved compared with the combustion of coal and oil in the traditional era of the energy supply chain. Cleaner resources, such as photovoltaics, are used as carriers, which not only reduces carbon emissions in the production process, but also improves the energy conversion efficiency, making the converted energy easier to recycle [52–55].

For energy transmission and distribution, the new energy supply chain pays more attention to the reserve of energy that is not needed yet. The continuous improvement in the layout of energy transmission and distribution stations, and the energy transmission and distribution network that crosses the whole country, greatly improve the timely transmission of energy resources, such as electric power and heat energy. The continuous improvement and innovation of energy storage technology also reduce the consumption of energy resources [56].

For energy consumption, the efficient and low-carbon operation of the front-end link of the new energy supply chain is followed by the end consumer and its demand satisfaction rate has been greatly improved compared with the traditional energy supply chain.

## 6. Discussion

### 6.1. Level of Energy Supply

In order to stabilize the guaranteed supply of fossil energy and realize incremental replacement to renewable energy, in terms of energy supply, we should speed up the reduction in the consumption of coal and non-fossil energy, ensure a basic supply of fossil energy, such as coal, and gradually replace it with renewable energy. At the same time, we should focus on innovative development of renewable energy development technologies to reduce the dependence on fossil-energy imports [57].

### 6.2. Level of Energy Production and Treatment

To improve conversion efficiency and reduce carbon emissions, we should innovate energy conversion and carbon-treatment technologies. The core manufacturing enterprises at the node of energy production and treatment need to constantly improve the energy conversion process, transform the related equipment and pay attention to the energy recovery process at the end of energy treatment. In the manufacturing process, the efficiency of energy conversion can be increased and the waste of energy can be reduced [58].

Considering the cost of comprehensive production and low-carbon transition, we should establish a mechanism of interest balance. The government needs to control the energy price in a unified way and, on the basis of a comprehensive consideration of the

costs of all nodes in the energy supply chain, establish a price mechanism that is acceptable to end consumers and profitable for the operators of energy resources.

### 6.3. Level of Energy Transmission and Distribution

It is important to build a nationwide transmission and distribution network to ensure timely energy transmission. In order to improve the overall performance of the Chinese energy supply chain, it is necessary to rationally plan the construction of energy transmission and distribution equipment and the scheduling of energy resources according to the regional distribution of the Chinese energy demand, considering the radiation range of transmission and distribution equipment. At the same time, we must develop energy-storage technologies and establish a safe and reliable energy-storage system [59].

### 6.4. Level of Energy Consumption

For improving the "double-carbon" awareness of terminal consumers and promoting the electrification of terminal electricity, it is necessary for the government and local governments to publicize the "double-carbon" goals and relevant national policies, so as to improve end consumers' awareness of "double-carbon" energy consumption and energy conservation and increase the consumption ratio of clean energy to a certain extent [60].

Conducting a reasonable "double-carbon" assessment to grasp the relationship between energy demand growth and green development, we must establish and perfect "double-carbon" policies, correctly handle the relationship between them and economic development, prevent "double-carbon" from restricting production and satisfy current energy needs while reducing carbon emissions, so as to achieve the goal of sustainable development [61].

### 7. Conclusions

In light of the rigid requirements of the "double-carbon" goals, the energy industry is facing higher carbon emission standards while promoting the development of the energy industry. How to accurately evaluate the comprehensive performance of the energy supply chain at this stage and improve the green sustainability of the energy supply chain while ensuring the overall operation efficiency of the energy supply chain has become an important measure for the energy industry achieving its "double-carbon" goals. Taking the energy supply chain as the specific object of study and adopting the analytic hierarchy process to construct an energy-supply-chain performance evaluation system, we evaluated the Chinese energy supply chain performance over the past 10 years based on AHP and a three-stage DEA analysis method, then noted some ways to improve and finally provided a strategy for promoting efficient "double-carbon" transformation. To sum up, the research conclusions of this paper are:

(1)   Based on the analysis of three-stage DEA and after excluding environmental variables, we found that the comprehensive efficiency of the Chinese energy supply chain showed a trend of increasing year on year on the whole, but there was a downward trend from 2010 to 2013; the comprehensive technical efficiency and comprehensive scale efficiency reached the lowest value in 2013 and began to rise after 2013. Compared with other years, 2019 can be considered the efficiency frontier in the development of the Chinese energy industry. Looking back on the development process of the Chinese energy industry management system, China established a sound and systematic energy management system combining professional supervision and comprehensive management in 2013. In 2013, the coal industry entered a period of structural optimization and various policies have been issued to support the use of clean energy and curb the consumption of coal, including an action plan for the prevention and control of air pollution, issued by the State Council in 2013, and the 2014–2015 action plan for energy conservation, emission reduction and low-carbon development (GBF [2014] No. 23). The support of a reasonable energy industry management system and energy conservation and emission reduction policies was

conducive to the efficient operation of the Chinese energy supply chain, which also provided financial support and an efficiency guarantee to help it carry out a green and low-carbon transformation. From 2017 to 2018, the efficiency of the Chinese energy supply chain increased significantly and reached the efficiency frontier in 2019.

(2) Through data analysis, we found that the comprehensive performance of the energy supply chain has improved since 2016. In order to achieve the "double-carbon" goals, although the Chinese energy transmission and distribution infrastructure needed to be continuously improved and gradually expanded to cover the whole country in 2016 and the energy-storage technology had only entered the initial stage of commercialization, China had begun to introduce new environmentally friendly energy transmission and distribution infrastructure. We should innovate and develop double-carbon energy-storage technologies, realize the full marketization of new energy storage, increase the investment in clean raw materials at the source of energy production, develop clean energy conversion efficiency and so on. We also need to reduce the consumption and import of primary energy, such as coal and oil, improve production efficiency so as to increase the output of finished energy products and lay a solid foundation for achieving "double-carbon" goals.

At present, there are still many problems with the development of the Chinese energy industry. The development of fossil fuel energy has caused damage to the environment and led to challenges for reductions in greenhouse gas emissions. China is under a certain pressure to achieve its "carbon peaking" and "carbon neutralization" ("double-carbon") goals, so we must improve them in terms of energy conservation and emission reduction.

**Author Contributions:** Project administration, X.L.; resources, J.Y.; validation, W.Z.; writing—original draft, Y.S.; writing—review and editing, X.H. All authors have read and agreed to the published version of the manuscript.

**Funding:** This research was funded by the National Social Science Foundation of China (17BGL008, 20XGL016), a Jiangxi Education Reform Project (JXJG-20-85-1), a Jiangxi Humanities and Social Science Research Project (YS20107), a Science and Technology Project of Jiangxi Provincial Department of Education (GJJ207103, GJJ191584) and the Jiangxi University of Science and Technology (XZG-15-01-05).

**Institutional Review Board Statement:** Not applicable.

**Informed Consent Statement:** Not applicable.

**Data Availability Statement:** The study did not report any data.

**Conflicts of Interest:** The authors declare no conflict of interest.

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
