# Peer review of "A Comprehensive Performance Evaluation of Chinese Energy Supply Chain under “Double-Carbon” Goals Based on AHP and Three-Stage DEA"

_sustainability, doi:10.3390/su141610149_

Round 1
Reviewer 1 Report
This paper performed analysis of Chinese energy supply chain under the “double-carbon” goal by using AHP and DEA analysis. The paper cannot be considered to be published before addressing the following comments.
- Pleas avoid using abbreviations in the title.
- Please provide full name for the abbreviations (AHP, DEA, BBC, CCR, SFA, etc) and symbols in equations when they appear as the first time.
- Language needs to be improved.
- Please rearrange the figure1 and figure 2 to make sure the caption follows the figure.
- Table 1. Do scales 5 and 7 have the same meaning?
- Table 3. What is the definition of clean energy of number 12.
- Line 370. Please clarify how A was derived?
- Please clarify the sources of the data (Table 3 and the data used to produce Table 6, etc) used for the analysis.
- The conclusions should be improved. Point (1) is not a direct conclusion from the study presented in this paper. Point (2) is methodology instead of conclusion. Point (3) seems not directly supported by the analysis and is away from the analysis under “double-carbon” Goals which should be the focus. Point (4) is an outlook or a future work from this study.
Author Response
Your suggestions: This paper performed analysis of Chinese energy supply chain under the “double-carbon” goal by using AHP and DEA analysis. The paper cannot be considered to be published before addressing the following comments.
Thank you so much for your precise and helpful suggestions, I will try my best to revise it !!!

Reviewer 2 Report
This work is an important research paper dealing with the concept of carbon neutrality, which the entire world is focused on recently. This work specifically focuses on the energy supply chain aspect of the story and focuses on providing suggestions for the energy industry.
Literature review is fine. There are many articles focused on similar topics recently and thus, I feel more articles published in 2022 could be referenced regarding carbon neutrality researches.
About Materials and Methods section, refer to the journal guide regarding listing section, subsection and subsubsection one after the other without any text. Also I am not sure if the indentation should be that way (lines 128,129,130)
Again, not sure if 139,140,141 can be condensed into a single figure instead of listing two figure captions one after the other.
There are some minor English issues, a minor spell check may be required.
Author Response
This work is an important research paper dealing with the concept of carbon neutrality, which the entire world is focused on recently. This work specifically focuses on the energy supply chain aspect of the story and focuses on providing suggestions for the energy industry.
Literature review is fine. There are many articles focused on similar topics recently and thus, I feel more articles published in 2022 could be referenced regarding carbon neutrality researches.
Thank you so much for your precise and helpful suggestions, you are so nice and kind, I will try my best to revise it !!!

Round 2
Reviewer 1 Report
Authors made changes based on the reviewer's comments. The paper can be published in the current form. One minor suggestion:
Please provide the reference of China Energy Statistical Yearbook.
This manuscript is a resubmission of an earlier submission. The following is a list of the peer review reports and author responses from that submission.